# Fast Tensor-Based Multi-View Clustering with Anchor Probability Transition Matrix

## Abstract

Multi-view clustering effectively integrates information from multi-view data representations, yet current methods face key challenges. They often lack interpretability, obscuring how clusters are formed, and fail to fully leverage the complementary information across views, limiting clustering quality. Additionally, large-scale data introduces high computational demands, with traditional methods requiring extensive post-processing.To address these issues, we propose a novel Fast Tensor-Based Multi-View Clustering with Anchor Probability Transition Matrix ((FTMVC-APTM). By selecting anchor points and constructing bipartite similarity graphs, we can capture the relationships between data points and anchors in different views and reduce computational complexity. Through probability matrices, we efficiently transfer cluster labels from anchors to samples, generating membership matrices without the need for post-processing. We further assemble these membership matrices into a tensor and apply a Schatten $p$-norm constraint to exploit complementary information across views, ensuring consistency and robustness. To prevent trivial solutions and ensure well-defined clusters, we incorporate nuclear norm-based regularization. Extensive experiments on various datasets confirm the effectiveness and efficiency of our method.

## 1 Introduction

In recent years, multi-view clustering (MVC) has gained increasing importance in machine learning and data analysis. As data sources expand through various sensors, imaging technologies, and social media platforms, multi-view data has become widespread across many fields. Unlike single-view clustering, which may miss important patterns by focusing on only one data perspective, MVC integrates information from multiple views to uncover the true underlying structure of the data (Chao et al., 2021).

Current MVC approaches can be categorized into four main types: subspace learning, graph-based methods, co-training methods, and multi-kernel learning. Subspace learning reduces the data to lower dimensions, which helps in handling high-dimensional datasets. However, this approach may fail to capture complex relationships between different views (Zheng et al., 2023; Gao et al., 2020a). Graph-based methods, on the other hand, build similarity graphs and apply spectral clustering. While effective, these methods can be computationally expensive due to the graph construction and eigendecomposition steps involved (Wei et al., 2017; Yang et al., 2023). Co-training improves clustering by combining classifiers from different views, especially when these views provide complementary information (Jiang et al., 2013). Multi-kernel learning captures non-linear relationships across views by learning a combined kernel, integrating information from multiple data sources (Tzortzis & Likas, 2012).

Despite their strengths, several challenges limit the practical application of these methods. Many existing approaches follow a two-step process: first, learning a fusion graph or spectral embedding, and then performing clustering. This separation often results in suboptimal performance, as the two steps are not jointly optimized. Moreover, many methods require complex post-processing to generate the final cluster labels, which increases computational complexity, particularly for large datasets (Brbić & Kopriva, 2018; Li et al., 2019). The separation of steps and the additional overhead make these methods less scalable for real-world applications (Yu et al., 2023).

To address these computational complexity issues, anchor graph-based methods have been proposed. These methods reduce the graph size by selecting a smaller subset of points (anchors) to represent the original data. By constructing a bipartite graph between the data points and the anchors, these methods significantly lower the computational burden during graph construction (Li et al., 2015). Li et al. (2024c) introduced tensor-anchor graph factorization by combining the concepts of tensors and anchor points. Additionally, Feng et al. (2024) proposed a depth tensor factorization method, which builds on depth matrix factorization to mine deeper, hidden information embedded in the anchor graph tensor. However, these methods rely on anchor graph data instead of raw data and provide only limited improvement in computational efficiency (Li et al., 2023; 2024a). After selecting anchors, Yu et al. (2023) constructed a probabilistic bipartite graph using both original and anchor data to derive a consensus matrix directly from the anchor label matrix. However, this method neglects the complementary information between multi-view data, which affects the overall clustering performance.

To overcome these challenges, we propose a novel method called Fast Tensor-Based Multi-View Clustering with Anchor Probability Transition Matrix (FTMVC-APTM), which simplifies the process and improves efficiency by directly using a probability transition matrix to derive the membership matrix from the anchor label matrix, eliminating the need for complex post-processing.This approach significantly reduces computational overhead and streamlines the entire clustering process while maintaining interpretability. To prevent trivial solutions and ensure well-defined clusters, we apply nuclear norm regularization to the membership matrix. Additionally, we apply a Schatten $p$-norm regularization to the tensor formed by the membership matrices across different views, thereby fully utilizing the complementary information between views and greatly improving clustering performance. The main contributions of our work are as follows:

- We propose a novel approach using probability matrices to directly compute membership matrices, avoiding the need for complex post-processing and enhancing clustering interpretability. This simplification enhances clustering efficiency, particularly for large datasets.

- Our method incorporates both nuclear norm and Schatten $p$-norm regularization to ensure balanced and robust clustering results. The nuclear norm promotes clear clusters and prevents trivial solutions, while the Schatten $p$-norm handles varied data distributions and mitigates the impact of noisy views. These techniques contribute to high-quality clustering outcomes.

- We conduct extensive experiments on multiple datasets to demonstrate the effectiveness and efficiency of our method. Results show that our approach outperforms existing methods in terms of both clustering accuracy and computational speed, highlighting its practical value for real-world applications.

## 2 RELATED WORK

### 2.1 NUCLEAR NORM IN MULTI-VIEW CLUSTERING

In multi-view clustering, imbalanced sample allocation can lead to two extremes: overly concentrated clustering and overly dispersed clustering. In the case of overly concentrated clustering, all data points are assigned to a single cluster. This results in a cluster assignment matrix where one column has non-zero entries while the rest remain zero. Such a matrix structure reflects limited diversity in the clustering, as the model essentially identifies only one cluster, providing little insight into the underlying data structure. Conversely, in overly dispersed clustering, the data points are evenly spread across all clusters, leading to a matrix where each column has equal entries. This uniform distribution makes it hard to discern meaningful groupings because the clustering fails to differentiate between the data points based on their inherent similarities.

To address this issue, Yu et al. (2023) introduced the nuclear norm as a regularization term to promote a balanced distribution of samples across clusters. The nuclear norm $\|Y\|_*$, defined as the sum of the singular values of the matrix $Y$, helps prevent extreme cases of over-concentration or over-dispersion by encouraging a more evenly distributed clustering. Formally, the nuclear norm is

expressed as:

$$\|Y\|_* = \mathrm{Tr}\left(\sqrt{Y^TY}\right) = \sum_{i=1}^{c} \sqrt{\rho_i(Y^TY)} \tag{1}$$

where $\rho_i(Y^TY)$ represents the $i$-th eigenvalue of the matrix $Y^TY$. Maximizing this norm helps avoid clustering outcomes that are too concentrated or too dispersed.

For example, in the case of overly concentrated clustering, the nuclear norm is low because the singular values reflect a lack of diversity in the cluster assignments. Conversely, in overly dispersed clustering, where each data point is equally distributed across clusters, the nuclear norm also remains low, as it fails to capture meaningful separations between groups.

The impact of the nuclear norm can be further understood through the following inequality:

$$\sum_{i=1}^{c} \sqrt{n_i} \le \sqrt{\sum_{i=1}^{c} n_i} = \sqrt{nc} \tag{2}$$

where $n_i$ denotes the number of samples in the $i$-th cluster. According to the Cauchy-Schwarz inequality, the nuclear norm reaches its maximum value when the number of samples in each cluster is equal, i.e., $n_1 = n_2 = \cdots = n_c = \frac{n}{c}$.

By maximizing the nuclear norm, clustering results are more balanced, ensuring that each sample is distinctly assigned to one of the clusters. This regularization method helps prevent trivial solutions and produces well-structured clustering outcomes that effectively capture the underlying structure of the data.

## 2.2 ANCHOR GRAPH-BASED MULTI-VIEW CLUSTERING

Anchor graph-based methods are widely adopted in multi-view clustering due to their ability to reduce computational complexity while maintaining performance. These methods select a smaller set of anchor points from the original data, constructing an $n \times m$ anchor graph that improves efficiency, especially in large datasets (Li et al., 2023). The concept of anchor points in multi-view clustering was first introduced by Liu et al. (2010), laying the groundwork for later advancements. Building on this, Li et al. (2015) proposed methods that replace the original data matrix with an anchor graph for each view and apply spectral clustering.

Further developments have expanded the use of anchor graphs in more sophisticated ways. Li et al. (2024c) introduced tensor-anchor graph factorization, which combines tensor structures with anchor points to capture more complex multi-view relationships. This method leverages both tensors and anchor points to enhance the clustering process. Li et al. (2023) proposed a depth tensor factorization method, building on matrix factorization techniques to uncover deeper, hidden information within anchor graph tensors. While this approach improves the ability to capture underlying data structures, its computational efficiency remains suboptimal when compared to other methods that more effectively leverage multi-view data.However, the reliance on anchor graphs rather than raw data provides only limited gains in computational efficiency. Yu et al. (2023) introduced a probabilistic bipartite graph by combining original and anchor data to directly derive a consensus matrix from the anchor label matrix. Although this method reduces computational complexity by using anchor points, it fails to fully exploit the complementary information between different views, which can limit overall clustering performance.

## 3 PROPOSED SCHEME

In this section, we introduce the motivation behind our proposed scheme, the detailed formulation of the objective function, and the optimization strategy employed to solve the problem. The notations used in the scheme are summarized in Table 1. Throughout this paper, matrices are denoted by bold uppercase letters (e.g., $X$), vectors by bold lowercase letters (e.g., $a$), and tensors by bold uppercase letters (e.g., $\mathcal{F}$).

## 3.1 MOTIVATION AND OBJECTIVES

Multi-view clustering aims to enhance clustering accuracy and robustness by leveraging complementary information from multiple data representations. However, many existing methods lack interpretability, making it difficult to understand how clusters are formed, especially when dealing with complex datasets. In addition, traditional methods often suffer from high computational complexity and require extensive post-processing, particularly for large-scale data.

To address these challenges, we propose a method that uses probability transition matrices combined with anchor label matrices to directly generate membership matrices. This approach not only simplifies the clustering process but also provides more straightforward and interpretable results by clearly showing how the anchor points relate to the final clusters, eliminating the need for complex post-processing.

Our method begins by selecting anchor points for each view from the original data matrix $\boldsymbol{X}^v \in \mathbb{R}^{n \times p_v}$, where $n$ is the number of data points and $p_v$ is the dimensionality of the $v$-th view. The anchor points $\boldsymbol{U}^v \in \mathbb{R}^{m \times d_v}$, with $m \ll n$, are a subset of representative points that capture the data distribution in a more compact form, thereby reducing computational complexity. By selecting a smaller set of anchors, we efficiently approximate the full dataset while retaining its structural properties.

Next, using the method in Nie et al. (2023), we construct bipartite similarity graphs that map the relationships between the data points in $\boldsymbol{X}^v$ and the anchor points in $\boldsymbol{U}^v$. The bipartite graph is characterized by the similarity matrix $\boldsymbol{B}^v \in \mathbb{R}^{n \times m}$, which encodes the relationships between the $n$ data points and the $m$ anchors for each view. Specifically, the bipartite graph is constructed as follows:

$$b_{ij} = \begin{cases} \frac{d(i,k+1)-d(i,j)}{kd(i,k+1)-\sum_{j=1}^{k} d(i,j)} & \forall j \in \Phi_i \\ 0 & j \notin \Phi_i \end{cases} \tag{3}$$

Here, $b_{ij}$ represents the similarity between the data point $x_i$ and the anchor point $u_j$, where $\Phi_i$ contains the indices of the $k$ nearest anchors of $x_i$, and $d(i,j)$ denotes the distance between $x_i$ and $u_j$. This approach ensures that the matrix $\boldsymbol{B}^v$ captures the probability transition between the data points and anchor points for each view.

To formalize, let $\boldsymbol{B}^v \in \mathbb{R}^{n \times m}$ denote the probability transition matrix for the $v$-th view, where $n$ is the number of data points and $m$ is the number of anchor points. The entries of $\boldsymbol{B}^v$ represent the probability of each data point being associated with each anchor point. We also define the anchor assignment matrix $\boldsymbol{Z}^v \in \mathbb{R}^{m \times c}$, where $c$ is the number of clusters. The entries of $\boldsymbol{Z}^v$ indicate the assignment of anchor points to clusters.

By directly transferring the labels from the anchor points to the samples, we define the membership matrix for the $v$-th view as:

$$\begin{aligned} \boldsymbol{F}^v &= \boldsymbol{B}^v \boldsymbol{Z}^v \\ \text{s.t.} \quad & \boldsymbol{Z}^v 1 = 1, \quad \boldsymbol{Z}^v \geq 0, \quad \boldsymbol{F}^v 1 = 1, \quad \boldsymbol{F}^v \geq 0, \end{aligned} \tag{4}$$

where $\boldsymbol{F}^v \in \mathbb{R}^{n \times c}$ represents the probability of each data point belonging to each cluster. We enforce the constraints $\boldsymbol{F}^v 1 = 1$ and $\boldsymbol{F}^v \geq 0$, ensuring that the cluster affiliations are valid probability distributions, which are non-negative and sum to one for each data point.

To avoid trivial solutions, as described in Section 2.1, we impose a nuclear norm constraint on the affiliation matrix $\boldsymbol{F}^v$. The nuclear norm encourages a clear separation of clusters by maximizing the rank of the affiliation matrix, ensuring that samples are well-distributed across clusters. This prevents scenarios where the clustering process results in overly concentrated or dispersed clusters, promoting a balanced allocation of samples and avoiding trivial solutions. The overall optimization problem can be formulated as follows:

$$\begin{aligned} \min_{\boldsymbol{Z}^v, \boldsymbol{F}^v} \sum_{v=1}^{V} & \left( \|\boldsymbol{B}^v \boldsymbol{Z}^v - \boldsymbol{F}^v\|_F^2 - \lambda \|\boldsymbol{F}^v\|_* \right) \\ \text{s.t.} \quad & \boldsymbol{Z}^v 1 = 1, \quad \boldsymbol{Z}^v \geq 0, \quad \boldsymbol{F}^v 1 = 1, \quad \boldsymbol{F}^v \geq 0, \end{aligned} \tag{5}$$

In to effectively integrate the complementary information from all the views, we form a tensor $\mathcal{F}$ of the membership matrices of each view in the same way as in Li et al. (2024c). Schatten $p$-norm is applied to the entire tensor, capturing the interactions and complementary information across views. The tensor is a tensor of the members of the view, and apply the Schatten $p$-norm (Gao et al., 2020b):

$$\min_{\boldsymbol{Z}^v, \mathcal{F}} \sum_{v=1}^{V} \left( \|\boldsymbol{B}^v \boldsymbol{Z}^v - \boldsymbol{F}^v\|_F^2 - \lambda \|\boldsymbol{F}^v\|_* \right) + \beta \|\mathcal{F}\|_{\omega, Sp}^p \tag{6}$$
$$\text{s.t.} \quad \boldsymbol{Z}^v 1 = 1, \quad \boldsymbol{Z}^v \geq 0, \quad \boldsymbol{F}^v 1 = 1, \quad \boldsymbol{F}^v \geq 0,$$

Here, $\beta$ is a parameter that controls the balance between global consistency and individual reconstruction accuracy, fostering a coherent yet flexible integration of multiple views.

Table 1: **Notations and Descriptions**

| Notation | Description |
|---|---|
| $\boldsymbol{X}^v \in \mathbb{R}^{n \times p_v}$ | Data matrix for the $v$-th view, where $n$ is the number of samples and $p_v$ is the dimension of the feature space in the $v$-th view |
| $\boldsymbol{B}^v \in \mathbb{R}^{n \times m}$ | Probability transition matrix for the $v$-th view, representing the relationship between data points and anchor points, where $m$ is the number of anchor points. |
| $\boldsymbol{Z}^v \in \mathbb{R}^{m \times c}$ | Anchor label matrix for the $v$-th view, where $c$ is the number of clusters |
| $\boldsymbol{F}^v \in \mathbb{R}^{n \times c}$ | Membership matrix for the $v$-th view, indicating the probability of each sample belonging to each cluster |
| $\mathcal{F} \in \mathbb{R}^{n \times c \times V}$ | Tensor consisting of $\boldsymbol{F}^v$ matrices from all $V$ views |
| $\mathcal{J}, \mathcal{W} \in \mathbb{R}^{n \times c \times V}$ | Auxiliary tensor variables used in the optimization process |
| $\rho$ | Penalty parameter |
| $\lambda, \beta$ | Regularization parameters |

## 3.2 OPTIMIZATION FRAMEWORK

To solve the optimization problem in Eq. equation 6, we introduce auxiliary variables $\mathcal{J}$ and Lagrange multipliers $\mathcal{W}$, with the dimensions of $\mathcal{J}, \mathcal{W} \in \mathbb{R}^{n \times c \times V}$, matching those of the membership tensor $\mathcal{F}$. These variables allow us to transform the constrained problem into an unconstrained one that can be solved iteratively using the Augmented Lagrange Multiplier (ALM) method.

The overall optimization problem is reformulated as:

$$\min_{\boldsymbol{Z}^v, \boldsymbol{F}^v, \mathcal{J}, \mathcal{W}} \sum_{v=1}^{V} \left( \|\boldsymbol{B}^v \boldsymbol{Z}^v - \boldsymbol{F}^v\|_F^2 - \lambda \|\boldsymbol{F}^v\|_* \right) + \beta \|\mathcal{J}\|_{\omega, Sp}^p + \frac{\rho}{2} \|\mathcal{F} - \mathcal{J} + \frac{\mathcal{W}}{\rho}\|_F^2 \tag{7}$$
$$\text{s.t.} \quad \boldsymbol{Z}^v 1 = 1, \quad \boldsymbol{Z}^v \geq 0, \quad \boldsymbol{F}^v 1 = 1, \quad \boldsymbol{F}^v \geq 0,$$

In this reformulation, $\mathcal{J}$ represents the auxiliary variable, and $\mathcal{W}$ represents the Lagrange multipliers. The penalty parameter $\rho$ controls the convergence of the ALM method. The optimization process is iteratively carried out until convergence, with each step involving updates to the variables $\boldsymbol{F}^v, \boldsymbol{Z}^v, \mathcal{J}$, and $\mathcal{W}$.

In the following, we describe the optimization process. For each variable, we optimize it while fixing the others, iterating through all variables until convergence.

**Optimization of $\boldsymbol{F}^v$:** After fixing the other variables, the optimization problem 7 for $\boldsymbol{F}^v$ is as follows:

$$\min_{\boldsymbol{F}^v} \sum_{v=1}^{V} \left( \|\boldsymbol{B}^v \boldsymbol{Z}^v - \boldsymbol{F}^v\|_F^2 - \lambda \|\boldsymbol{F}^v\|_* \right) + \frac{\rho}{2} \|\mathcal{F} - \mathcal{J} + \frac{\mathcal{W}}{\rho}\|_F^2 \tag{8}$$
$$\text{s.t.} \quad \boldsymbol{F}^v 1 = 1, \quad \boldsymbol{F}^v \geq 0$$

The Frobenius norm term in equation 8 can be expanded as:

$$\|\boldsymbol{B}^v\boldsymbol{Z}^v - \boldsymbol{F}^v\|_F^2 = \text{Tr}((\boldsymbol{F}^v)^T\boldsymbol{F}^v) - 2\text{Tr}((\boldsymbol{F}^v)^T\boldsymbol{B}^v\boldsymbol{Z}^v) + \text{Tr}((\boldsymbol{B}^v\boldsymbol{Z}^v)^T\boldsymbol{B}^v\boldsymbol{Z}^v) \tag{9}$$

The term $\text{Tr}((\boldsymbol{B}^v\boldsymbol{Z}^v)^T\boldsymbol{B}^v\boldsymbol{Z}^v)$ is constant and can be ignored during optimization. The nuclear norm term contributes a subgradient:

$$\boldsymbol{D}^v = \frac{\partial\|\boldsymbol{F}^v\|_*}{\partial\boldsymbol{F}^v} = \boldsymbol{F}^v((\boldsymbol{F}^v)^T\boldsymbol{F}^v)^{-\frac{1}{2}} \tag{10}$$

The ADMM penalty term is:

$$\frac{\rho}{2}\|\boldsymbol{\mathcal{F}}^v - \left(\boldsymbol{\mathcal{J}}^v - \frac{\boldsymbol{\mathcal{W}}^v}{\rho}\right)\|_F^2 = \frac{\rho}{2}\text{Tr}((\boldsymbol{F}^v)^T\boldsymbol{F}^v) - \rho\text{Tr}\left((\boldsymbol{F}^v)^T\left(\boldsymbol{\mathcal{J}}^v - \frac{\boldsymbol{\mathcal{W}}^v}{\rho}\right)\right)$$
$$+ \text{Tr}\left(\left(\boldsymbol{\mathcal{J}}^v - \frac{\boldsymbol{\mathcal{W}}^v}{\rho}\right)^T\left(\boldsymbol{\mathcal{J}}^v - \frac{\boldsymbol{\mathcal{W}}^v}{\rho}\right)\right) \tag{11}$$

Based on this, we can rewrite equation 8 as follows:

$$\min_{\boldsymbol{F}^v1=1,\boldsymbol{F}^v\geq0}\sum_{v=1}^V d^v(\|\boldsymbol{B}^v\boldsymbol{Z}^v - \boldsymbol{F}^v\|_F^2 - \lambda\text{Tr}((\boldsymbol{D}^v)^T\boldsymbol{F}^v)) + \frac{\rho}{2}\|\boldsymbol{\mathcal{F}} - \boldsymbol{\mathcal{J}} + \frac{\boldsymbol{\mathcal{W}}}{\rho}\|_F^2$$

$$\Leftrightarrow \min_{\boldsymbol{F}^v1=1,\boldsymbol{F}^v\geq0} d^v\text{Tr}(\boldsymbol{F}^{vT}\boldsymbol{F}^v - 2\boldsymbol{F}^{vT}\boldsymbol{B}^v\boldsymbol{Z}^v) - \lambda\text{Tr}(\boldsymbol{F}^{vT}\boldsymbol{D}^v) + \frac{\rho}{2}\text{Tr}(\boldsymbol{F}^{vT}\boldsymbol{F}^v)$$

$$- \rho\text{Tr}(\boldsymbol{F}^{vT}(\boldsymbol{\mathcal{J}}^v - \frac{\boldsymbol{\mathcal{W}}^v}{\rho}))$$

$$\Leftrightarrow \min_{\boldsymbol{F}^v1=1,\boldsymbol{F}^v\geq0}\text{Tr}((d^v + \rho)\boldsymbol{F}^{vT}\boldsymbol{F}^v - \boldsymbol{F}^{vT}(2d^v\boldsymbol{B}^v\boldsymbol{Z}^v + \rho(\boldsymbol{\mathcal{J}}^v - \frac{\boldsymbol{\mathcal{W}}^v}{\rho}))) - \lambda\text{Tr}(\boldsymbol{F}^{vT}\boldsymbol{D}^v)$$

$$\Leftrightarrow \min_{\boldsymbol{F}^v1=1,\boldsymbol{F}^v\geq0}\left\|\boldsymbol{F}^v - \frac{\boldsymbol{B}^v\boldsymbol{Z}^v + \frac{\lambda}{2}\boldsymbol{D}^v + \rho(\boldsymbol{\mathcal{J}}^v - \frac{\boldsymbol{\mathcal{W}}^v}{\rho})}{d^v + \rho}\right\|_F^2 \tag{12}$$

Problem 12 can be solved by the solution in Yu et al. (2023).

**Optimization of $\boldsymbol{Z}^v$:** After fixing the other variables, the optimization problem can be formulated as:

$$\min_{\boldsymbol{Z}^v1=1,\boldsymbol{Z}^v\geq0}\|\boldsymbol{B}^v\boldsymbol{Z}^v - \boldsymbol{F}^v\|_F^2 \tag{13}$$

This problem can be rewritten as:

$$\min\left\|[\mathbf{b}^v \quad \boldsymbol{B}^r]\begin{bmatrix}\mathbf{z}^v\\\boldsymbol{Z}^r\end{bmatrix} - \boldsymbol{F}^v\right\|_F^2$$
$$\Leftrightarrow \min\|\mathbf{b}^v\mathbf{z}^v + \boldsymbol{B}^r\boldsymbol{Z}^r - \boldsymbol{F}^v\|_F^2 \tag{14}$$
$$\Leftrightarrow \min\left\|z^v - \frac{(\boldsymbol{B}^r\boldsymbol{Z}^r - \boldsymbol{F}^v)^Tb^v}{(b^v)^Tb^v}\right\|_2^2$$

where $\mathbf{z}^v$ denotes the $i$-th row of $\boldsymbol{Z}^v$ and $\mathbf{b}^v$ denotes the $i$-th column of $\boldsymbol{B}^v$. Problem 14 is similar to Problem 12 can be solved by the solution in Yu et al. (2023).

**Optimization of $\boldsymbol{\mathcal{J}}^v$:** After fixing the other variables, the optimization problem 7 for $\boldsymbol{Z}^v$ is as follows:

$$\min_{\boldsymbol{\mathcal{J}}^v}\frac{\rho}{2}\|\boldsymbol{\mathcal{F}} - \boldsymbol{\mathcal{J}} + \frac{\boldsymbol{\mathcal{W}}}{\rho}\|_F^2 + \beta\|\boldsymbol{\mathcal{J}}\|_{\omega,Sp}^p$$
$$\text{s.t.} \quad \boldsymbol{\mathcal{J}}^v \geq 0 \tag{15}$$

after completing the square regarding $\boldsymbol{\mathcal{J}}$, we can deduce

$$\boldsymbol{\mathcal{J}}^* = \arg\min\frac{1}{2}\left\|\boldsymbol{\mathcal{H}} + \frac{\boldsymbol{\mathcal{Y}}_2}{\rho} - \boldsymbol{\mathcal{J}}\right\|_F^2 + \frac{\lambda}{\rho}\|\boldsymbol{\mathcal{J}}\|_{\boldsymbol{\mathcal{S}}_p}^p \tag{16}$$

Based on Zhao et al. (2024), the optimal solution for Eq.15 is given by:

$$\mathcal{J}^* = \Gamma_{\frac{\beta}{\rho}} \left( \mathcal{F} + \frac{\mathcal{W}}{\rho} \right) \tag{17}$$

Here, $\Gamma_{\frac{\beta}{\rho}}$ is a generalized shrinkage operator that applies the Schatten $p$-norm regularization to the tensor $\mathcal{F} + \frac{\mathcal{W}}{\rho}$. This operator helps control the rank of $\mathcal{J}$, improving the robustness of the solution.

**Update of $\mathcal{W}^v$:**

Finally, the Lagrange multipliers $\mathcal{W}^v$ are updated to ensure consistency between $\mathcal{J}^v$ and $\boldsymbol{F}^v$:

$$\mathcal{W}^v = \mathcal{W}^v + \rho(\boldsymbol{F}^v - \mathcal{J}^v) \tag{18}$$

The optimization procedure is outlined in Algorithm 1.

---

**Algorithm 1** Fast Tensor-Based Multi-View Clustering with Anchor Probability Transition Matrix (FTMVC-APTM)

---

**input:** Multi-view data $\{\boldsymbol{X}^V\}_{v=1}^V$, anchor number $c$, regularization parameters $\lambda$, $\beta$
**output:** Clustering labels for each sample
 1: Initialize variables $\boldsymbol{Z}^v$, $\boldsymbol{F}^v$, $\mathcal{J}^v$, $\mathcal{W}^v$, $\mu = 1.6$
 2: Compute anchor graph matrix $\boldsymbol{B}^v$ for each view
 3: **while** not converged **do**
 4:   **for** each view $i = 1$ to $V$ **do**
 5:     Update $\boldsymbol{F}^v$ using Eq. 12
 6:     Update $\mathcal{J}^v$ using Eq. 17
 7:     Update $\boldsymbol{Z}^v$ using Eq. 14
 8:     Update $\mathcal{W}^v$ using Eq. 18
 9:     Update $\rho = \min(\mu\rho, 10^{13})$
10:   **end for**
11: **end while**
12: Compute final clustering labels based on $\boldsymbol{F} = \sum_{v=1}^V \boldsymbol{F}^v / V$
13: **return** Clustering result(The position of the largest element in each row of the indicator matrix is the label of the corresponding sample).

---

### 3.3 COMPLEXITY ANALYSIS

The proposed FTMVC-APTM algorithm consists of several stages: (1) Compute the similar bipartite graph $\boldsymbol{B}^v$;(2) updating the anchor label matrix $\boldsymbol{Z}^v$; (3) updating the membership matrix $\boldsymbol{F}^v$ for each view and the auxiliary variable $\mathcal{J}$;

$\boldsymbol{B}^v$ needs to be computed only once and its computational complexity is $O(nmV)$.In the update phase, let the number of iterations be $t$. The first step is to update the anchor label matrix $\boldsymbol{Z}^v$. This step has a complexity of $O(nmcV)$, where $n$ is the number of data points and $m$ is the number of anchor points. Next, the update of the membership matrix $\boldsymbol{F}^v$ requires matrix multiplications, resulting in a complexity of $O(nm^2cV)$. The auxiliary variable $\mathcal{J}$, used for the Schatten $p$-norm regularization, adds an additional complexity of $O(2Vnclog(Vc) + V^2cn)$, due to the computations involving the norm regularization.Considering that $V, c$ are small constants, $m \ll n$,thus the computational complexity of the scheme MVCt should be $O(t(nm^2cV + nmcV + V^2cn))$, which is proportional to the magnitude of $n$, showing the efficiency of the FTMVC-APTM.

The appendix includes a comparison of the computational complexity and running time of the FTMVC-APTM with the comparison methods to demonstrate the efficiency of our method again.

## 4 EXPERIMENTS

### 4.1 DATASET

We evaluate the performance of the proposed method on eight widely adapted multi-view learning benchmark datasets, which are **Yale**(Yale University, 2001), **BBCSport**(Greene & Cunning-

ham, 2006), **MNIST** (Deng, 2012), **Uci-digit**, **NGs**(Hussain et al., 2010),**WebKB**(Blum & Mitchell, 1998),**MSRC**(Winn & Jojic, 2005) and **SentencesNYU v2 (RGB-D)**(Silberman et al., 2012). Detailed information on dataset specifications is provided in Table 2.

## 4.2 COMPARISON METHODS

We have selected nine representative multi-view clustering (MVC) algorithms for comparison: **GMC** (Wang et al., 2019) and **MvLRSSC** (Brbić & Kopriva, 2018) are graph-based methods that use graph structures to capture relationships between views. **MVC-DMF-PA** (Zhang et al., 2021) applies matrix factorization, while **MVC-DNTF** and **Orth-NTF** (Li et al., 2024b) utilize tensor factorization with anchor points to reduce computational complexity. **FastMICE** (Huang et al., 2023) and **FPMVS-CAG** (Wang et al., 2021) also rely on anchor points to accelerate clustering. Finally, **RMSL** (Li et al., 2019) and **MVFCAG** (Zhao et al., 2024) incorporate probabilistic models, with **MVFCAG** using probabilistic matrices to refine clustering.

## 4.3 EXPERIMENTAL SETUP

All experiments were executed on a desktop with an Intel(R) Core(TM) i5-13400 CPU and 32 GB of RAM, using MATLAB 2023a. Data normalization was performed as a preprocessing step for all datasets to ensure consistent input quality. We assessed the clustering quality using Accuracy (ACC), Normalized Mutual Information (NMI), and Purity (PUR). Each experiment was replicated 5 times, and the best result was selected to avoid the impact of randomness.

Table 2: Dataset specifications

| Dataset | Views | Dimension | Samples | Clusters |
|---------|-------|-----------|---------|----------|
| BBCSport | 2 | 3283/3183 | 544 | 5 |
| Yale | 2 | 1024/4096 | 165 | 11 |
| Minst4 | 3 | 30/9/30 | 4000 | 4 |
| Uci-digit | 3 | 216/76/64 | 2000 | 10 |
| NGs | 3 | 2000/2000/2000 | 500 | 5 |
| WebKB | 2 | 1840/3000 | 1051 | 2 |
| MSRC | 5 | 24/576/512/256/254 | 210 | 7 |
| RGB-D | 2 | 2048/300 | 1449 | 13 |

Table 3: Clustering performance comparison in terms of ACC, NMI, and PUR on Yale, BBCSport, Minst4, and Uci-digit datasets.

| Datasets | Yale | | | BBCSport | | | MNIST | | | Uci-digit | | |
|----------|------|------|------|----------|------|------|-------|------|------|-----------|------|------|
| Metrics | ACC | NMI | PUR | ACC | NMI | PUR | ACC | NMI | PUR | ACC | NMI | PUR |
| FastMICE | 65.46 | 66.06 | 47.04 | 41.91 | 46.00 | 7.90 | 48.77 | 33.56 | 47.57 | 84.05 | 86.25 | 85.95 |
| MvLRSSC | 58.79 | 39.20 | 66.09 | 76.63 | 72.36 | 76.63 | 54.52 | 24.67 | 43.25 | 80.36 | 76.78 | 81.89 |
| RMSL | 78.78 | 78.23 | 79.39 | 76.63 | 72.36 | 76.63 | 54.92 | 25.03 | 46.32 | 51.90 | 52.05 | 55.95 |
| GMC | 54.55 | 62.44 | 54.55 | 80.70 | 76.00 | 79.43 | 88.17 | 73.81 | 79.14 | 83.90 | 87.41 | 86.35 |
| FPMVS-CAG | 50.31 | 59.32 | 51.52 | 42.10 | 15.09 | 51.84 | 65.15 | 11.91 | 40.92 | 75.30 | 75.87 | 75.35 |
| MVFCAG | 51.52 | 55.47 | 40.38 | 38.79 | 9.51 | 38.68 | 91.87 | 79.82 | 85.76 | 84.01 | 85.09 | 83.48 |
| MVC-DMF-PA | 15.75 | 16.10 | 20.00 | 73.34 | 52.68 | 76.28 | 59.04 | 39.05 | 49.73 | 73.20 | 75.26 | 70.44 |
| Orth-NTF | 78.18 | 81.90 | 80.00 | 89.15 | 79.49 | 89.52 | 94.07 | 85.65 | 89.39 | 93.75 | 90.27 | 89.35 |
| MVC-DNTF | 84.24 | 86.39 | 82.42 | 98.05 | 87.85 | 94.85 | 95.15 | 86.87 | 91.00 | 89.10 | 85.06 | 82.49 |
| OURS | **97.57** | **96.95** | **95.15** | **98.34** | **94.87** | **96.78** | **98.75** | **95.38** | **97.54** | **98.15** | **96.19** | **96.40** |

## 4.4 EXPERIMENT RESULTS

The clustering performance of our proposed method was evaluated against nine representative multi-view clustering (MVC) algorithms across several benchmark datasets. We report the results in terms of Accuracy (ACC), Normalized Mutual Information (NMI), and Purity (PUR). The experimental results are shown in Table 3 and Table 4, where the best results are bolded and the second-best results are underlined.

Table 4: Clustering performance comparison in terms of ACC, NMI, and PUR on NGs, WebKB, MSRC, and RGB-D datasets.

| Datasets | NGs | | | WebKB | | | MSRC | | | RGB-D | | |
|---|---|---|---|---|---|---|---|---|---|---|---|---|
| Metrics | ACC | NMI | PUR | ACC | NMI | PUR | ACC | NMI | PUR | ACC | NMI | PUR |
| FastMICE | 38.40 | 48.00 | 26.63 | 95.62 | 94.63 | 0.66 | 86.67 | 86.67 | 77.73 | 41.81 | 32.61 | 49.53 |
| MvLRSSC | 90.26 | 88.82 | 91.72 | 92.58 | 58.19 | 92.58 | 78.57 | 68.55 | 78.57 | 39.00 | 32.40 | 50.59 |
| RMSL | 9.60 | 86.11 | 94.60 | 60.42 | 1.93 | 78.12 | 27.62 | 8.18 | 31.90 | 12.63 | 2.85 | 26.98 |
| GMC | 97.80 | 92.93 | 97.80 | 84.02 | 25.78 | 84.02 | 24.29 | 6.91 | 26.19 | 40.23 | 33.06 | 46.51 |
| FPMVS-CAG | 73.80 | 59.23 | 73.80 | 94.96 | 69.91 | 94.96 | 42.86 | 37.68 | 42.86 | 34.50 | 38.73 | 45.47 |
| MVFCAG | 27.60 | 6.01 | 36.52 | 79.16 | 0.695 | 73.94 | 90.74 | 81.84 | 90.74 | 33.33 | 23.68 | 24.76 |
| MVC-DMF-PA | 86.80 | 80.27 | 86.80 | 89.43 | 50.89 | 89.43 | 91.43 | 85.36 | 91.43 | 16.83 | 72.25 | 33.12 |
| Orth-NTF | 95.40 | 89.73 | 95.40 | 96.57 | 73.25 | 96.57 | 98.09 | 96.02 | 98.09 | 59.07 | 65.78 | 75.56 |
| MVC-DNTF | 97.60 | 93.73 | 97.60 | 95.81 | 71.55 | 95.81 | 97.61 | 95.30 | 97.61 | 63.21 | 71.28 | **82.95** |
| OURS | **99.40** | **97.91** | **98.80** | **100.00** | **100.00** | **100.00** | **99.04** | **97.84** | **98.09** | **78.60** | **82.88** | 81.66 |

In Table 3, our method demonstrates superior clustering performance on most datasets. For example, on the Yale, BBCSport, MNIST, and Uci-digit datasets, our proposed method achieves ACC values of 97.57%, 98.34%, 98.75%, and 98.15%, respectively, significantly outperforming other methods. The NMI and PUR metrics also reflect a similar trend, where our method consistently achieves higher scores, illustrating the effectiveness of our approach in accurately capturing multi-view data characteristics.

Similarly, in Table 2, our method continues to lead on the NGs, WebKB, MSRC, and RGB-D datasets, obtaining almost perfect results in terms of ACC and NMI. Specifically, on the WebKB dataset, our method achieves 100% in all three metrics, showcasing its robustness and ability to handle diverse datasets. Even for more challenging datasets, such as RGB-D, our method still shows a clear advantage over the other approaches, achieving ACC of 78.60% and NMI of 82.88%, which are considerably higher than those achieved by the other methods.

The overall results show that our method not only effectively utilizes the complementary information between multiple views, achieves good interpretability and efficiency, but also maintains quite impressive clustering results. As a result, it achieves remarkable clustering accuracy across various types of datasets, further proving the robustness and versatility of the proposed approach.

## 4.5 PARAMETER ANALYSIS

We conducted experiments to evaluate the influence of key parameters on our clustering method. Specifically, we analyzed how varying the Schatten $p$-norm parameters $\beta$ and $p$, as well as the anchor rate and the nuclear norm regularization parameter $\lambda$, affects clustering performance.

As shown in Figure 1, the clustering accuracy remains relatively stable across different values of $\beta$ and $p$, demonstrating the robustness of our method to these parameters. However, we observe that the optimal performance is generally achieved when $p$ is between 0.4 and 0.6.

In Figure 2, we examine the impact of the anchor rate and $\lambda$ on clustering accuracy. The results indicate that the accuracy is not significantly affected by changes in the anchor rate, highlighting the robustness of our method to this parameter. For the BBCSport, MSRC, and Yale datasets, the optimal performance is achieved when $\lambda$ is between 0.5 and 1.0. In contrast, the WebKB dataset achieves optimal results when $\lambda$ is between 1.75 and 2.25.

## 4.6 ABLATION STUDY

To evaluate the impact of the nuclear norm and Schatten $p$-norm constraints in our proposed method, we performed ablation experiments under four different settings. In case 1, only the nuclear norm is applied, while in case 2, only the Schatten $p$-norm is applied. We compare these cases to a baseline where neither constraint is used and to the full model where both constraints are incorporated.

The results, as shown in Table 5, indicate that without either constraint (baseline), the model yields poor performance across all datasets, with accuracy ranging from 36.99% to 41.72%. When only the Schatten $p$-norm is applied (case 2), the accuracy improves slightly for certain datasets, such as Yale and RGB-D, but remains low overall. This suggests that while the Schatten $p$-norm helps cap-

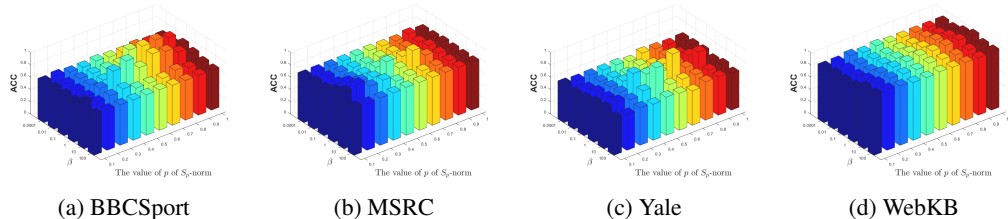

Figure 1: The influence of the Schatten $p$-norm and $\beta$ on clustering results for the BBCSport, MSRC, Yale, and WebKB datasets.

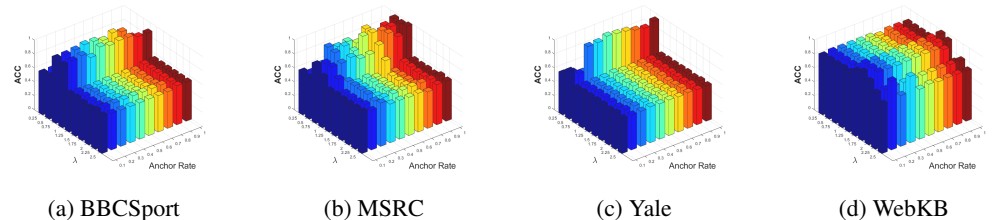

Figure 2: The influence of the anchor rate and $\lambda$ on clustering results for the BBCSport, Sonar, Yale, and RGB-D datasets.

Table 5: ACC(%) of ablation experiments

| | | Datasets | | | |
|---|---|---|---|---|---|
| case1 | case2 | MSRC | Yale | RGB-D | BBCSport |
| × | × | 39.52 | 38.18 | 36.99 | 41.72 |
| × | ✓ | 46.66 | 64.24 | 41.75 | 39.52 |
| ✓ | × | 76.19 | 52.72 | 42.09 | 61.76 |
| ✓ | ✓ | 99.04 | 97.57 | 78.60 | 98.34 |

ture complementary information across views, it struggles to produce coherent and well-structured clustering results on its own. In contrast, applying only the nuclear norm (case 1) significantly boosts performance across most datasets, with accuracy reaching 76.19% on MSRC and 61.76% on BBCSport, highlighting its importance in ensuring robust and non-trivial clustering structures. Finally, the full model, combining both constraints, delivers the best performance on all datasets, with accuracies close to or above 97%, demonstrating the synergy of using both regularization terms.

## 5 CONCLUSION

In this paper, we proposed a Fast Tensor-Based Multi-View Clustering with Anchor Probability Transition Matrix (FTMVC-APTM), which simplifies the clustering process by directly using anchor-based probability transition matrices. This eliminates the need for complex post-processing and improves computational efficiency. By integrating nuclear norm and Schatten $p$-norm regularization, the method ensures well-defined clusters while fully utilizing complementary information from multiple views. Extensive experiments show that FTMVC-APTM consistently outperforms existing methods in terms of both accuracy and speed, particularly on large datasets. Future work may focus on further optimizing the method towards a parameter-free approach, reducing the reliance on manual parameter tuning and improving its adaptability across diverse datasets. In conclusion, FTMVC-APTM provides an efficient and scalable solution to multi-view clustering, making it suitable for various practical scenarios.

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
