# OpenReview forum: "Fast Tensor-Based Multi-View Clustering with Anchor Probability Transition Matrix"
_ICLR.cc/2025/Conference — Submitted to ICLR 2025_

### Official Review · Reviewer_JsSs · 2024-10-21

**Soundness:** 1
**Presentation:** 2
**Contribution:** 2
**Rating:** 3
**Confidence:** 5

**Summary:**

This paper proposed a Fast Tensor-Based Multi-View Clustering with Anchor Probability Transition Matrix ((FTMVC-APTM) to address some key challenges in multi-view clustering like lacking interpretability, and high computational complexity from large-scale data. Extensive experiments on various datasets are conducted to demonstrate the effectiveness and efficiency.

**Strengths:**

A good framework for this paper.

**Weaknesses:**

1.	Innovative may not be enough for such a conference, this work simply combines a lot of work, for example, nuclear norm and Schatten p-norm regularization are both very common regular terms, and the authors don't discuss in depth why they use these two items， for example why Schatten p-norm regularization, there are many new low-rank tensor norm [1][2].
2.	The article is poorly expressed, for example, whether the author employs Schatten p-norm or weighted tensor Schatten p-norm. the introduction states the Schatten p-norm, but Eq.6 uses the weighted tensor Schatten p-norm in [3]. These are two completely different concepts. If you use the weighted tensor Schatten p-norm, how did you determine the weight values for the different views?
3.	This work states “fast tensor-based multi-view clustering”, but the dataset is only 4k in size and there is no runtime comparison, which is hard to believe!
4.	The author states “Each experiment was replicated 5 times”, so why do the results in Table 3 not include variance?
5.	In Figure 2, the performance always reaches best when anchor rate=1, which means the anchor is useless, and the complexity is also O(n^2logn), This result proves that the work proposed by the authors is not valid, At least it contradicts the author's “fast” statement.


[1] Guo J, Sun Y, Gao J, et al. Logarithmic Schatten-$ p $ p Norm Minimization for Tensorial Multi-View Subspace Clustering[J]. IEEE Transactions on Pattern Analysis and Machine Intelligence, 2022, 45(3): 3396-3410.

[2] Ji, Jintian, and Songhe Feng. "Anchor structure regularization induced multi-view subspace clustering via enhanced tensor rank minimization." Proceedings of the IEEE/CVF International Conference on Computer Vision. 2023.

[3] Gao, Quanxue, et al. "Enhanced tensor RPCA and its application." IEEE transactions on pattern analysis and machine intelligence 43.6 (2020): 2133-2140.

**Questions:**

See Weaknesses.

---

### Official Review · Reviewer_gJfc · 2024-10-27

**Soundness:** 3
**Presentation:** 3
**Contribution:** 2
**Rating:** 5
**Confidence:** 4

**Summary:**

This paper proposes a new multi-view clustering method called Fast Tensor Multi-view Clustering Based on Anchor Probability Transformation Matrix (FTMVC-APTM). The method of directly calculating the membership matrix using the probability matrix avoids complex post-processing and enhances clustering interpretability. The nuclear norm and Schatten p-norm regularization are introduced to ensure the balance and robustness of the clustering results.

**Strengths:**

1. The proposed method is clearly explained and easy to understand.

2. This paper carefully analyzes the computational complexity of the method and illustrates the potential advantages of FTMVC-APTM in data scale expansion.

3. Experimental results on eight multi-view datasets demonstrate its effectiveness.

**Weaknesses:**

1. The main contributions of this paper is to combine the anchor probability transformation matrix and the Schatten p-norm regularization of the multi-view tensor structure. However, these ideas are not new in the field of multi-view clustering, and the combination of anchor selection, tensor structure and probability matrix has been applied in some methods[1][2].
[1] Nie, Feiping, et al. "Fast clustering with anchor guidance." IEEE Transactions on Pattern Analysis and Machine Intelligence (2023).
[2] Yu, Weizhong, et al. "Multi-View Fuzzy Clustering Based on Anchor Graph." IEEE Transactions on Fuzzy Systems (2023).

2. This paper lacks ablation experiments on the key design of using probability matrix to calculate membership matrix. Given that this method is a core contribution of FTMVC-APTM, conducting relevant ablation experiments will help evaluate the actual impact of this strategy on the model performance.

3. Although this paper demonstrates the superior performance of FTMVC-APTM on multi-view datasets, the scale of these datasets is relatively limited (the number of samples ranges from a few hundred to a few thousand), which fails to fully verify the performance of the method on large-scale data. It is recommended to supplement the experiments on larger datasets, such as the YTF dataset and the Caltech dataset.

4. It is recommended that the authors appropriately increase the visualization results of clustering to help readers more intuitively understand the performance and clustering structure of the proposed FTMVC-APTM method.

**Questions:**

1. This paper claims that this method is more interpretable than other complex multi-view clustering methods. Specifically, how does the membership matrix generated by the probability matrix help explain the final clustering structure?

---

### Official Review · Reviewer_d6Mp · 2024-11-01

**Soundness:** 3
**Presentation:** 3
**Contribution:** 2
**Rating:** 5
**Confidence:** 4

**Summary:**

In this paper, a tensor-based method is proposed to solve the MVC problem. The authors propose a simple and efficient method and verify the rationality and superiority of the method through experimental results.

**Strengths:**

1.	This paper is well organized.
2.	This paper implements an exploration of fast tensor clustering.
3.	The proposed methodology is somewhat enlightening.

**Weaknesses:**

I have some concerns about the paper, as follows:

1. This paper is also limited by its innovative. Affiliation matrix is not a new method, it has been widely used[1,2]. Tensor Schatten-p norm[3,4] are also a common way to deal with low rank. So, the innovation made by the authors is more in the sense of incremental.

[1] Zhao, J. B., & Lu, G. F. (2022). Clean and robust affinity matrix learning for multi-view clustering. Applied Intelligence, 52(14), 15899-15915.

[2] Li, X., Zhang, H., Wang, R., & Nie, F. (2020). Multiview clustering: A scalable and parameter-free bipartite graph fusion method. IEEE Transactions on Pattern Analysis and Machine Intelligence, 44(1), 330-344.

[3] Xie, Y., Gu, S., Liu, Y., Zuo, W., Zhang, W., & Zhang, L. (2016). Weighted Schatten
-norm minimization for image denoising and background subtraction. IEEE transactions on image processing, 25(10), 4842-4857.

[4] Li, X., Ren, Z., Sun, Q., & Xu, Z. (2023). Auto-weighted tensor schatten p-norm for robust multi-view graph clustering. Pattern Recognition, 134, 109083.

2. The experimental results in this paper are inadequate. For example, the authors emphasize that their method enhances the interpretability of clustering. However, this needs to be verified experimentally. The superior performance of clustering alone may not provide effective support.

3. In addition, the authors emphasize that their method requires only linear complexity and has a fast computational speed. However, the sample size of the dataset used is small, and I suggest the authors to increase their experiments on large-scale datasets such as AwA[5] or Youtube[6].

[5] https://cvml.ista.ac.at/AwA/

[6] https://www.cs.tau.ac.il/~wolf/ytfaces/

**Questions:**

1.	The sample size of the existing dataset is small and the authors should increase the experiments on large-scale datasets.
2.	The available experimental results are not sufficient to support the authors' opinion. I suggest the authors to add some visualization or other experiments.
3.	Compared to existing methods, the authors' innovation is unclear. I suggest that the authors should carefully consider the motivation and contributions.

---

### Official Review · Reviewer_K6zb · 2024-11-02

**Soundness:** 3
**Presentation:** 3
**Contribution:** 2
**Rating:** 3
**Confidence:** 5

**Summary:**

In this paper, the authors propose a Fast Tensor-Based Multi-View Clustering with Anchor Probability Transition Matrix ((FTMVC-APTM) method. Within this model, to reduce the computational complexity, the relationships between data points and the selected anchors in different views are captured, and recorded by the bipartite similarity graphs. Based on these probability graphs,  the cluster labels from anchors to samples are transferred, and the membership matrices can be obtained without the need for post-processing. To further exploit complementary information across views, the membership matrices are stacked into a tensor and contrained by a Schatten p-norm.

**Strengths:**

S1. This paper is easy to follow, and the idea is straightforward.

S2. The introduction about the framework is clear, and the equations are well presented.

**Weaknesses:**

W1. Overall, the idea of this paper is straightforward and clear. However, the novelty of FTMVC-APTM is limited, and the presented motivations/problems have been referred and solved.

W2. The used datasets are too small, and the experiments provided in this paper are not convincing to show the superiority of the proposed method.

W3. The running time comparison experiment is missing.

W4. More recent fast multi-view clustering methods should be introduced and compared in the experiments.

**Questions:**

See weakness.

---

### Meta-Review · Area_Chair_VXZj · 2024-12-16

**Metareview:**

In this paper, the authors directly compute the membership matrix using the probability matrix to avoid complex post-processing and enhance the clustering interpretability. The nuclear norm and Schatten p-norm regularization are introduced to ensure the consistency and robustness of the clustering results. The membership matrices are stacked into a tensor to further exploit complementary information across views. Extensive experiments on various datasets confirm the effectiveness and efficiency.





All reviewers give negative scores.  Its novelty is limited.  Affiliation matrix and Tensor Schatten-p norm are fairly common.  The experimental results   are inadequate, and the running time comparison experiment is missing. Moreover, some recent works should be introduced and compared in experiments. The scale of utilized datasets is relatively limited, failing to verify the performance of proposed method on large-scale data. The expression needs further refinement. Also, there is no any rebuttal.

**Additional Comments On Reviewer Discussion:**

All reviewers give negative scores.  Its novelty is limited.  Affiliation matrix and Tensor Schatten-p norm are fairly common.  The experimental results   are inadequate, and the running time comparison experiment is missing. Moreover, some recent works should be introduced and compared in experiments. The scale of utilized datasets is relatively limited, failing to verify the performance of proposed method on large-scale data. The expression needs further refinement.

---

### Decision · Program_Chairs · 2025-01-22

Reject